# Convolutional Neural-Network-Based Reverse-Time Migration with Multiple Reflections

**DOI:** 10.3390/s23084012

**Published:** 2023-04-15

**Authors:** Shang Huang, Daniel Trad

**Affiliations:** Department of Geoscience, University of Calgary, 2500 University Drive NW, Calgary, AB T2N 1N4, Canada; daniel.trad@ucalgary.ca

**Keywords:** convolutional neural network, reverse-time migration, surface-related multiples

## Abstract

Reverse-time migration (RTM) has the advantage that it can handle steep dipping structures and offer high-resolution images of the complex subsurface. Nevertheless, there are some limitations to the chosen initial model, aperture illumination and computation efficiency. RTM has a strong dependency on the initial velocity model. The RTM result image will perform poorly if the input background velocity model is inaccurate. One solution is to apply least-squares reverse-time migration (LSRTM), which updates the reflectivity and suppresses artifacts through iterations. However, the output resolution still depends heavily on the input and accuracy of the velocity model, even more than for standard RTM. For the aperture limitation, RTM with multiple reflections (RTMM) is instrumental in improving the illumination but will generate crosstalks because of the interference between different orders of multiples. We proposed a method based on a convolutional neural network (CNN) that behaves like a filter applying the inverse of the Hessian. This approach can learn patterns representing the relation between the reflectivity obtained through RTMM and the true reflectivity obtained from velocity models through a residual U-Net with an identity mapping. Once trained, this neural network can be used to enhance the quality of RTMM images. Numerical experiments show that RTMM-CNN can recover major structures and thin layers with higher resolution and improved accuracy compared with the RTM-CNN method. Additionally, the proposed method demonstrates a significant degree of generalizability across diverse geology models, encompassing complex thin layers, salt bodies, folds, and faults. Moreover, The computational efficiency of the method is demonstrated by its lower computational cost compared with LSRTM.

## 1. Introduction

Reverse-time migration (RTM) [1,2,3,4] can handle steep geologic structure flanks and lateral velocity variations. However, it suffers from coherent or incoherent artifacts in diving and backscattered waves, as well as low resolution and illumination for deep structure when given insufficient source-receiver offsets. A solution to suppress the artifact issue is applying least-squares reverse-time migration (LSRTM) [5], which uses RTM as the forward modeling and inverse engine to minimize amplitude differences between observed data and predicted data and updates the reflectivity iteratively. Extensive research on least-squares imaging such as compressive sensing [6], uncertainty quantification [7], sparsity constraints [8], curvelet-domain sparse constraint [9], and multiplicative Cauchy constraint [10] help to improve imaging.

Although LSRTM improves illumination with respect to RTM, it still has a limited aperture problem, since it uses only primary reflections. Multiple migration used in imaging [11] and the RTM (RTMM) [12,13,14,15] can help to broaden the subsurface illumination and refine the accuracy and resolution.

Another way to enhance imaging quality is through deep learning. Many researchers have worked on this approach and addressed accuracy improvement and artifact suppression in seismic processing. For example, ground roll attenuation [16,17], seismic inversion applications [18,19,20,21], transfer learning applications in modeling and imaging [22], and generative neural networks in inverse problems [23,24] proposed to use the multilayered convolutional neural network (CNN) [25,26,27] as the solution to the problem of sparse least-squares migration (LSM) to suppress coherent and incoherent noise in migration results. For deep learning in RTM and least-squares RTM, [28] proposed LSRTM with the adaptive moment estimation in the frequency domain; Ref. [29] applied a generative adversarial network on RTM images with a velocity attribute conditioner to estimate the inverse of the Hessian and match with least-squares migrated images; Ref. [30] also used CNNs on dip-angle domain elastic reverse-time migration to improve image quality; Ref. [31] introduced the idea on minibatch LSRTM, and Torres and Sacchi [32,33] used blocks of residual CNN on LSRTM with a preconditioned conjugate gradient least-squares algorithm (CGLS) to enhance image resolution; Ref. [34] used deep learning for accelerating prestack correlative LSRTM. These methods mitigate artifacts and foster resolution by training a machine learning network.

Exploiting the two facts that multiple reflections can enhance the imaging bandwidth and a convolutional neural network (CNN) can learn the lithologic structure from different feature maps, we propose a CNN-based RTM with the multiple reflections energy method (RTMM-CNN). In this approach, we use a U-Net [35] acting as a filter to learn the reflection boundaries from the RTMM results, and we also make the filter learn the mapping of multiple energy. We use the U-Net-based RTM image as the baseline model without adding multiple energy (RTM-CNN). Models have two components: preconditioning and fine-tuning. The preconditioning constrains the parameter range in the fine-tuned models, improving the image quality. Results show that the proposed method can obtain the reflectivity prediction with extended illumination, refined structural boundaries, high accuracy, and enhanced resolution.

## 2. Methods

Before diving into the theory part, some basic geophysical variables need to be explained. Seismic waves, generated from the simulation of sources on the surface, are extrapolated downwards into the subsurface. When they encounter some discontinuities of physical properties, reflection and transmission waves are generated on the interfaces.

These discontinuities are called seismic impedance discontinuities. They are the product of subsurface velocity and density and are components of the reflectivity equation to generate wavefield perturbations.
(1)Z=ρv
where ρ is the density of rock and *v* denotes the velocity of that rock.

The goal of seismic imaging is to estimate, for each point of the subsurface, a parameter called reflection coefficient, or reflectivity. This parameter indicates the ratio of waves reflected and transmitted through a discontinuity media, compared with incident waves. For an acoustic normal incident wave situation, the expression of the reflection coefficient is
(2)$$R=\frac{Z_{2}-Z_{1}}{Z_{2}+Z_{1}}$$
where $Z_{1}$ and $Z_{2}$ mean the impedance of the first and second medium, respectively. Thus, the target of seismic migration and imaging is to obtain the approximation of reflectivity and interpret subsurface structures by using collected wavefields.

### 2.1. Four Scenarios in Reverse-Time Migration

Reverse-time migration involves two wavefields. The source wavefield produces the wave propagation from shots, and the receiver wavefield reproduces the time-reverse wave propagation from receivers (which is acquired data). Since the cross-correlation of these two wavefields generates the desired reflectivity, it is essential to make them physically consistent. The shot wavefield will be limited mainly by our knowledge of velocities. Since velocities are usually not known at a level of detail to create internal reflections, we usually deal with what is called the Born wavefield (no internal reflections). This wavefield may or may not contain surface multiples depending on how we perform modeling (for example, using or not using an absorbing boundary condition on the surface). For the receiver wavefield, however, there is a limitation not only on the velocities (the same as for the source wavefield) but also on what the data contain (were internal or external multiples attenuated before migration?). There are several possibilities for the relationship between these wavefields. For this paper, we can separate the following four scenarios, although only scenarios 1, 3, and 4 are critical for the following discussions:Scenario 1: Smooth or background velocity input to RTM and absorbing boundary conditions on the surface. Multiples were attenuated from data before migration. This scenario is the typical case in real data applications. The forward wavefield will be Born modeling (no internal multiples).Scenario 2: True velocity input to RTM and data without multiples. This scenario is not practical, because true velocities are not known at that level of detail. There is an inconsistency between forward wavefields (which will have all multiples) and reverse wavefields (which will have some attenuated multiples and some generated during backward propagation). We leave this scenario out of the discussion.Scenario 3: Smooth velocity input to RTM and data with multiples. This scenario is our goal, because it is close to reality (only the background model is known), and data will contain multiples (unless attenuated explicitly). These multiples will provide additional illumination for RTM. Nonetheless, there is an inconsistency between wavefields that has to be addressed.Scenario 4: True velocity input to RTM and data with multiples. As for this scenario, it is also hard to perform in reality, but it has an ideal result that combines high-frequency bandwidth and multiple reflections, which makes migrated results with high resolution and insight for subsurface structure interpretation. We try to build a neural network on the basis of scenario 3 and make the result close to scenario 4.

From Figure 1, we can see that RTM in scenario 1 can adequately recover the structure of a relatively simple thrust model. As expected, the illumination is stronger on the shallow part of the thrust structure than on the deeper one. Significant shallow depth illumination is the typical RTM result expected in practice. Although there are some difficulties to achieve in practical conditions, the RTM of scenario 4 is an ideal result. The thrust structure is estimated with high resolution and accuracy, and the illumination is very good for shallow and deep reflectors. The RTM of scenario 3 seems somewhat better than in scenario 1 but worse than in scenario 4. The illumination is not inadequate for the deeper geological structure. However, the more complicated the model, the more deterioration of the image appears, as the crosstalk will introduce more artifacts. If the thrust model had many reflectors in between, the crosstalk would be significantly visible, and the image would be poor. The situation above is easy to obtain in practical applications if the multiples are not removed from the data before migration.

### 2.2. Wavefield Inconsistency

As mentioned above, an RTM with a smooth velocity and data with multiples (scenario 3) will have inconsistency between wavefields. The forward wavefield will consist of traveltimes from primaries, surface multiples, and internal multiples during the imaging process. However, the reverse wavefield will contain primaries and attenuated multiples due to the smooth background velocity model, which does not generate internal reflections. Since the velocity model lacks the information to unravel the traveltimes for multiples properly, the receiver wavefield will incorrectly cross-correlate with the source wavefield, and crosstalk noise will occur [5,12].

To mitigate the traveltime mismatch, previous work, for example, (Schuster [36,37]) and Jiang et al. [38], proposed using modified Green’s functions to migrate multiples. The method above constructs the Kirchhoff imaging condition for multiples by combining traveltime picked from a shifted source wavelet and obtained data, which is based on Fermat’s principle. It assumes that the traveltimes of a lower-order event can be picked (or windowed in the prestack data) in order to image a higher-order event. Another essential factor is that the multiples’ energy must be sufficient for picking.

To alleviate the wavefield inconsistency and traveltime mismatch in the reverse-time migration result, our proposed method lets the neural network learn the traveltime mismatch and correct the inconsistency iteratively during training. Certainly, this is not a trivial assumption. Like other deep learning applications, it is impossible to provide a justification or proof that this would be the case, and experimentation is critical. The benefit is that we can take advantage of all the receiver’s information, including primaries, multiples, or crosstalks (if they help improve reflection coefficient information). Thus, the extended illumination brought from multiple reflections can help us improve the subsurface image result. A smoothed reflectivity model from background velocity and corresponding RTM outcome with multiple energy are considered the input channels of this neural network. In other words, we want to try to obtain high-resolution results of scenario 4 from scenario 3.

### 2.3. RTM with Multiples

The workflow of reverse-time migration is given by a forward- and reversed-time propagation of source and receiver wavefields, respectively, followed by an imaging condition. As multiple reflections are considered in the RTM process, free surface boundaries need to add to the top of velocity models.

Similar to Liu et al. [12], using primaries P(x,z,t), we apply the total observations (P(x,z,t)+M(x,z,t)) as the virtual source to generate multiples $M^{\prime }\left(x,z,t\right)$, where M(x,z,t) represents internal multiples generated from the first forward modeling. Then, multiples $M^{\prime }\left(x,z,t\right)$ have the total wavefields, including primaries and surface and internal multiples. During the imaging process, those virtual-source wavefields will be forward-extrapolated into the subsurface as $P_{F}\left(x,z,t\right)+M_{F}\left(x,z,t\right)$. The newly acquired data $M^{\prime }\left(x,z,t\right)$ will be backpropagated into the subsurface and considered as our reversed-time receiver wavefields $M_{B}^{\prime }\left(x,z,t\right)$.

A zero-lag cross-correlation imaging condition based on the virtual-source and receiver wavefields Liu et al. [12] can be applied:(3)$$I\left(x,z\right)=\sum_{t=1}^{t_{max}}\left(P_{F}\left(x,z,t\right)+M_{F}\left(x,z,t\right)\right)*M_{B}^{\prime }\left(x,z,t\right)$$

Even though the imaging condition generates crosstalks, the trained neural network can learn patterns and features from the relationship between the migrated result and the true reflectivity model. In that case, the neural network can exploit the benefits of multiple energy and mitigate artifacts in the image.

Figure 2 shows the Pluto migration images with respect to a background reflectivity by applying RTM (scenario 1, Figure 2c), RTMM (scenario 3, Figure 2d), and RTMM with true band-limited reflectivity (scenario 4, Figure 2e), respectively. The migrated image using multiple energy (Figure 2d) can help extend horizontal-layer illumination, recover deep thin-layer structures and provide lateral continuity structural information. It is beneficial for a neural network to identify reflection events using multiple reflections in the migration. In the next section, we see that reflectivity predictions can be improved by using migration velocity models as constraints introduced into the network in a secondary channel.

### 2.4. A U-Net-Based RTM with Multiples

Given a particular type of input in deep learning, we train a multilayer network to predict the desired outcome. A series of weights are calculated to map the inputs to the desired output during training. Although, essentially, this is just a geometrical mapping, the transformation contains both linear (the weights) and nonlinear elements (the activation functions), which, added to a large number of weights, have the potential to take into account many complex effects. By choosing the types of inputs and the desired outputs (known as labels), we can make the network learn any particular mapping we need. There are many limitations on what this type of geometrical transformation can learn, but more importantly, there are limitations on the generality of this mapping. In general, it is difficult to say whether a particular mapping will succeed, and we often have to rely on numerical experiments to achieve a conclusion. In this paper, we propose to use a U-Net (Figure 3) to map the outcomes of a reverse-time migration obtained from data with multiples and a migration velocity model (smooth) to a well-resolved image (RTMM-CNN). The reflectivity obtained from the migration velocity is incorporated as a physical constraint to provide low-frequency information to the network.

U-Net [35] is an encoder–decoder approach commonly used for image segmentation (pixel classification), but we utilize it here for a regression problem. In this paper, we develop a U-Net (Figure 3) with additional multilayer convolutional blocks and skip connections to learn from residuals and patterns in the data. Convolutional blocks are used for capturing detailed input features. For example, they help distinguish signals and noise from images with multiple reflection information. The network downsamples the input data into small sizes for the encoder part. It reduces its dimensionality to learn key features of different reflectors from RTMM images, smoothed initial reflectivity, and accurate reflectivity labels. Then, these subsurface key features are upsampled to the original dimensions by transposed convolutions. Additional skip connections work as identity mapping because the signal could be directly propagated from one unit to any other unit [39]. These identity shortcut connections help to smooth key feature propagation and strengthen the training result with weak constraints. For the output layer, a linear activation function is used for obtaining positive and negative prediction values, which obey the nature of reflectivity amplitude.

The U-Net provides a mechanism to design a prediction filter from our training data (the RTM images and additional support channels) to the labels (the simulated reflectivity obtained from true velocities). This U-Net operator acts similarly to an image domain LSRTM, but the inverse of the Hessian is calculated not from inverse filtering but by training. In comparison, LSRTM in the image domain yields a high-definition image by removing the effect of the Hessian from the migrated image. The calculation of the inverse of the Hessian filter can be calculated with different methods [40,41,42,43,44,45], but for simplicity, we can summarize as follows:(4)$$m^{*}=\arg\min_{m}\{\frac{1}{2}||\Gamma m-m_{mig}{\left|\right|}_{2}^{2}\}.$$

A formal solution to Equation (Equation 4) is
(5)$$m^{*}=\Gamma^{-1}m_{mig}=\Gamma^{-1}\left(L^{T}d\right),$$
(6)$$\Gamma =L^{T}L,$$
where $\Gamma^{-1}$ is the inverse Hessian, $L^{T}$ is the adjoint operator, and d represents the observed seismic data. From Equation (Equation 5), we can find that prediction $m^{*}$ is generated from the deblurring process of the first migrated image $m_{mig}$ by the inverse of the Hessian matrix Γ.

Similarly, a neural network, U-Net in our case, can be used as an approximate inverse Hessian [29,32] to determine the imaging result. The benefit is that there is no need to compute the expensive inverse Hessian operator. The Hessian contains the effects of limited acquisition aperture, uneven illumination, and band-limited source wavelets. These effects compromise the goal of obtaining a true-amplitude and high-resolution reflectivity [45]. The feed-forward procedure in our proposed method for a multilayer CNN is $\Gamma_{unet}$, and the solution can be determined as follows, depending on the different scenarios:(7)$$\mathrm{Workflow}1:m_{pred_{1}}=\Gamma_{unet\_fine\_tuned\_workflow2}\left(m_{rtm\_scenario1},m_{smooth}\right),$$
(8)$$\mathrm{Workflow}2:m_{pred_{2}}=\Gamma_{unet\_workflow2}\left(m_{rtm\_scenario2},m_{true}\right),$$
(9)$$\mathrm{Workflow}3:m_{pred_{3}}=\Gamma_{unet\_fine\_tuned\_workflow4}\left(m_{rtmm\_scenario3},m_{smooth}\right),$$
(10)$$\mathrm{Workflow}4:m_{pred_{4}}=\Gamma_{unet\_workflow4}\left(m_{rtmm\_scenario4},m_{true}\right),$$
where $m_{rtm}$ means RTM image, $m_{rtmm}$ is the RTMM image, and $m_{pred}$ represents the output reflectivity coefficient prediction. The subscripts after $m_{rtm}$ and $m_{rtmm}$ correspond to the different scenarios mentioned before. For instance, $m_{rtm\_scenario1}$ represents the RTM image from scenario 1. The neural networks used in the workflows are set individually depending on the preconditioning or final imaging demand. For example, the neural network model generated from workflow 2 ($\Gamma_{unet\_workflow2}$) is used for preconditioning, because its inputs are the true reflectivity and the RTM image from scenario 2. Then, this saved model is treated as a pretrained model for workflow 1. After fine-tuning, workflow 1 will obtain a final training model $\Gamma_{unet\_fine\_tuned\_workflow2}$. A detailed explanation for decisions about preconditioning and final imaging models is delineated in a later section. As for $m_{true}$ and $m_{smooth}$, the former denotes the true band-limited reflectivity, which is used as our labels during training; the latter is an initial reflectivity calculated from the background velocity used for migration. This reflectivity contains only low-frequency information, similar to what migration normally uses, but uses it as an additional input channel, providing a supplementary constraint for the network. The workflows correspond to scenarios 1 to 4, mentioned previously.

For a detailed U-Net architecture (Table 1 and Table 2), there are 45 layers for encoding and 44 layers for decoding, respectively. In the contracting path, each convolutional block has three convolutional layers for the first four blocks. The last two blocks contain two convolutional layers. After each block, the maxpooling layer, with a size of 2 by 2 cells, halves the image’s size and increases the neural network’s depth. Table 1 indicates that, at the end of the encoding part, the image size is reduced from 256 × 256 × 2 to 4 × 4 × 512 by using the convolutional blocks and maxpooling layers. The number of channels increases from 2 to 512. On the other side, in the expansive path, extracted features are upscaled by transposed convolutional layers and back to the image’s original size. Following transposed convolution, the resized image is concatenated with an image from the contracting path sharing the same size. Skip connection combines previous image information and makes a stable and accurate prediction. Before outputting the prediction, another skip connection layer is added to obtain a precise result.

Mean Squared Error (MSE)

As estimating the reflectivity coefficient from a seismic migration profile with an initial reflectivity model is a regression problem, a mean squared error (MSE) loss is used to evaluate the model performance and calculate the gradient:(11)$$\mathrm{MSE}=\frac{1}{n}\sum_{i=1}^{n}{\left(m_{pred}^{i}-m_{true}^{i}\right)}^{2},$$
where *n* is the total number of samples, $m_{pred}$ is derived from the workflows above (Equations (Equation 7)–(Equation 10)), and $m_{true}$ denotes the true reflectivity model.

Peak signal-to-noise ratio (PSNR)

A peak signal-to-noise ratio (PSNR) is used to evaluate the model performance:(12)$$\mathrm{PSNR}=20\;\log_{10}\left(\frac{MAX_{I}}{\sqrt{MSE}}\right),$$
where $MAX_{I}$ denotes the maximum possible pixel value of the image, and MSE is the mean squared error based on the Equation (Equation 11).

### 2.5. Neural Network Plan for Four Workflows

In this section, we introduce in detail four workflows that are defined in the previous section. To train the neural network and make it learn patterns from both accurate and smoothed inputs, workflows 1 and 3 are fine-tuned based on the neural networks obtained from workflows 2 and 4. The networks trained by workflows 2 and 4 act as initialization and regularization constraints. The pretraining process can act as a regularizer [46] to introduce a helpful prior and implicitly minimize the appropriate parameters’ range for the next steps of fine-tuning training. For high-level abstraction learning in a deep architecture, the regularizer imposes some constraints on the parameters to direct the minima where the cost function seeks. As in workflows 2 and 4, true reflectivity is used as one input channel. This helps to reduce the neural network parameter space and provides fine-tuned neural networks in workflows 1 and 3 with an initial model to train on. Even though the input channel changes to a smoothed background reflectivity, the neural network will learn the critical reflector information. Furthermore, the pretrained models prevent networks 1 and 3 from creating new reflectors or artifacts not present in their inputs. Because the migrations from sharp velocities used in workflows 2 and 4 are impossible in practice, these networks cannot be used directly during inference. Instead, they help to initialize and constrain the other networks for optimization. A detailed description is shown in Figure 4, where reflectivity and RTM/RTMM images are the input for training the neural network. This neural network plan is similar to the idea of ensemble learning [47,48,49], which combines several learning algorithms to solve the same problem for obtaining a better prediction. This paper’s workflows share the same training neural network structure but with different training inputs. The difference is that workflows 2 and 4 are first trained using the true band-limited reflectivity and corresponding RTM/RTMM images obtained from scenarios 2 and 4, respectively. Then, workflows 1 and 3 use the pretrained models R2 and R4 to fine-tune the neural network given on a smoothed input, whose RTM and RTMM images are generated from scenarios 1 and 3 and produce the updated models R1 and R3. Model R1, meaning the network trained from workflow 1, is then used to predict our baseline model, which is a result that we can easily obtain but want to improve.

Note that the smoothed input now is the reflectivity calculated from the background smoothed velocity. The reason for smoothing the input is that the neural network tolerates slight incorrect velocity errors more. After learning patterns from smoothed inputs, the neural network can distinguish reflectivity events from crosstalk or artifacts. This process can mitigate unexpected noise from migrated images and result in high accuracy and resolution. The comparison between different model outputs and performance is illustrated in detail in the numerical result section. We expect that, in general, models R1 and R3 will produce better images than R2 and R4. Furthermore, we expect that R3 can take advantage of multiple reflections’ wide illumination and information to predict an improved reflectivity.

Although scenario 4’s RTMM image is fed into workflow 4 to train a preconditioned model R4, we want to let the fine-tuned model R3 learn and predict an accurate image close to scenario 4’s output. To clarify the whole process, let us recall the definitions of scenario 4, workflow 4 and model 4: scenario 4 uses true velocity to obtain the RTM with surface multiples; workflow 4 uses the true reflectivity and output, as we would have in scenario 4. In this workflow, a U-Net trains the input with two channels: true reflectivity and RTMM images from scenario 4. The model R4 is then stored from workflow 4 and will be used as a pretrained model for workflow 3. After training, the model R3 will be a fine-tuned neural network obtained from workflow 3, which uses smooth reflectivity and RTMM images as input. The initial model for R3 was R4 (from workflow 4); therefore, there will be an improvement from model R4.

### 2.6. Train and Test Set

We chose a series of common velocity models for training: Sigsbee2b, Amoco, Pluto, BP2004, Marmousi I and II, and others we built arbitrarily. We generated synthetic data, migrated all these velocity models, and used their RTM/RTMM images as training data. We calculated their reflectivities from their true velocities as training labels and made them band-limited by convolving with a time-domain 25 Hz Ricker wavelet. The shots and receivers were located just below the surface with 160 and 16 m spacing, using a cell size of 8 m. The sources are Ricker wavelets with a 20 Hz dominant frequency. The total record time was variable and longer for the deeper salt models, with a maximum of 7.2 s for the Sigsbee2b case. For the modeling and migration, we used a fourth-order finite-difference method with a 15 Hz dominant frequency, implemented with CUDA for GPUs [50].

We employed data augmentation techniques such as image resizing and smoothing to increase the dataset instead of image rotation and flipping. Although very common in computer vision, these last two techniques are inappropriate in seismic imaging, because physical and geological principles constrain geophysical images. The vertical direction represents depth, while the horizontal direction denotes offset. Rotating or flipping the images would violate the fundamental principle that migration results are obtained from seismic wave propagation. We defined shots on the surface and simulated them to generate seismic waves that were then extrapolated into the subsurface. These waves were reflected and transmitted by subsurface structures, and the receivers on the surface generated shot records. The final step is to use these shot records to migrate reflections to their correct positions and create subsurface images. Therefore, we refrained from using rotation or flipping as a data augmentation method. Furthermore, no new data points were created in the input.

Our baseline model R1 is trained on workflow 1, corresponding to scenario 1, using smoothed reflectivity calculated from the background velocity and RTM images without multiples as the input channels. On the other hand, the proposed model R3 uses RTM images with multiple energy as one of the inputs. Before training, the RTM and RTMM images of scenarios 2 and 4 are divided randomly into 2700 spatial windows with 256 × 256 grid points. For example, the Pluto model has 601 × 1750 points. Suppose the random sequence numbers for horizontal distance and depth are 100 and 50, respectively. In that case, a chosen window should be located in the original model with offset numbers 100-356 and 50-306 points in depth, because the window size is 256 points in a square shape. Working in windows is not only practical for handling large images but also introduces a regularization effect since, if the predictions are correct, they should contain the same information where they overlap. If the predictions are not similar, the summation of predictions from different windows will reduce the resolution.

As for scenarios 1 and 3, RTM/RTMM images are separated into 2500 subwindows. The train and validation set ratio is 0.8:0.2. We chose these numbers because a large training data size can help generalization. The maximum number of iterations for each training model is limited to 200 using the Adam optimizer with a batch size of 64. The learning rate is reduced during iterations to avoid the solution falling into local minima.

Then, we test our neural networks on three examples: the Canadian Foothills, a 2D slice of the Overthrust, and the SEAM Phase 1 geology models. These examples were not used during the training to test generalization; that is, how the neural network performs on new data. In the next section, we show a detailed comparison between neural networks in different scenarios and situations.

## 3. Results and Discussions

This section tests predictions for the Canadian Foothills, Overthrust, and SEAM examples by independently working through workflows 1, 3, and 4. These examples test the neural networks’ capability for generalization. The spatial interval for each example is 8 m with an 0.8 ms time sampling rate. As discussed previously, we can use the prediction from model R4, the network trained from workflow 4, as a reference and a regularization network. This result corresponds to a neural network trained using true band-limited reflectivity and RTMM images. However, the results could be better, because the inputs for inference are migrations with wavefield inconsistency (data with multiples migrated with smooth velocity). The model R1 from workflow 1 is our baseline model trained on a smoothed input without multiple energy.

On the other hand, although workflow 3 is similar to workflow 1, utilizing a smoothed reflectivity input, model R3 has RTMM images as input instead of RTM images. After learning patterns from multiple reflections’ energy and smoothed inputs, model R3 can distinguish reflectivity events from crosstalk or artifacts. This process can mitigate migrated noise and improve resolution. Detailed numerical analysis and comparison is shown in the next part.

### 3.1. Example 1: Canadian Foothills

The Canadian Foothills example initially has 1000 × 1600, but we chose 768 × 1536 gridpoints for the neural network prediction. We simulated 78 shots and 795 receivers at the near surface, with 160 and 16 m spacing separately. Figure 5 shows the results of the neural network models R1, R3, and R4 predictions on workflows 1, 3, and 4, correspondingly. A smoothed reflectivity generated from the background velocity (Figure 5a) is the first input channel for models R1, R3, and R4. Note that the smooth reflectivity input (Figure 5a) is in its original value, which does not have high frequencies due to smoothing, but the amplitude will be scaled during testing. RTM image (Figure 5c) is set as the second input for model R1; on the other hand, RTM image with multiples (Figure 5d) is used as the second input channel for models R3 and R4. The migration of the Canadian Foothills model using multiples with smooth velocity (Figure 5d) has increased illumination and artifacts due to the wavefield inconsistency described earlier. The result for workflow 3 (model R3 (Figure 5f)) tries to correct for these inconsistencies and shows somewhat better lateral event continuity with artifact reduction in comparison with model R1 prediction (Figure 5e), which did not use multiples. For example, the shallow curvatures in the middle horizontal distance can be seen clearly in Figure 5f, with higher resolution and less noise (Table 3) in comparison with Figure 5e. Additionally, compared with the model R4 result (Figure 5g), which is set as our reference, model R3 (Figure 5f) can also give a more accurate prediction of geological structures with improved resolution, which is closer to the true band-limited reflectivity (Figure 5b) calculated directly from the velocity model.

The example was also tested with LSRTM, and after 15 iterations, the resulting image is displayed in Figure 5i. Compared with RTM and RTMM results, the LSRTM image provides additional high-resolution information about the reflectors, particularly the side curvature boundary between 10,000 and 12,000 m at a depth of approximately 2000 m. However, the computational cost of LSRTM is at least twice that of RTMM for one iteration, and it requires several hours to complete 15 iterations, even when using OpenMPI. This time is longer than required for RTMM calculation and neural network training. On the other hand, the model R3 result, which is a fast approximation of the LSRTM output, can recover most of the reflectors with noise suppression. Therefore, this result confirms that the proposed method achieves enhanced efficiency in reflectivity calculation compared with the LSRTM method.

In Figure 6, we compare the average amplitude spectrum of the results of the different networks. The reflectivity obtained from the smooth velocity model and used as an input channel (long dashed line) has lost low and high frequencies as expected, since it comes from a background velocity. Even though model R1 (point dashed line) can aid in recovering low frequencies between 0.002 and around 0.002 m${}^{-1}$, model R3 (solid line) predicts a broader frequency band and higher values after about 0.008 m${}^{-1}$. This observation indicates that model R3 takes advantage of true band-limited reflectivity on the low-frequency band, which is learned from the pretrained network R4 and the multiple energy from RTMM images on high frequencies. Thus, model R3 can predict and rebuild more information on low and high frequencies, promoting output resolution and accuracy.

When comparing the peak signal-to-noise ratio shown in Table 3, the model R3 application has the highest value of the three results, which means it has the most confidence in predictions compared with other outputs. Most of the structural information of the true band-limited reflectivity image is preserved.

Figure 7 contains the same information as Figure 5 but zoomed into a window at middle depth (red block No. 1 in Figure 5h). A normalization scaling is applied to make the smooth reflectivity input more visible. The migrated image with multiple energy (Figure 7d) shows more illumination than the regular (no-multiples) RTM image (Figure 7c). For instance, the small fault on the top right can be migrated with higher illumination in the RTMM image (Figure 7d). The prediction from model R3 (Figure 7f) shows fewer artifacts and enhanced resolution and accuracy, whereas the prediction from model R1 (Figure 7e) shows artifacts at around 2800 m in depth. The forecast from model R4 (Figure 7g) is similar to the smooth reflectivity input. It shows no improvements as expected, since the network from workflow four was pretrained using accurate inputs and cannot handle smoothing inputs.

Figure 8 shows a different (shallower) window (red block No. 2 in Figure 5h). The multiples used in the RTMM image (Figure 8d) predict workflow 3 (Figure 8f) better than the predictions from workflows 1 and 4 (Figure 8e,g). For example, the depth structure between 0 and 1000 m can be predicted with larger amplitude and accuracy in Figure 8f. Moreover, the model R3 result can have better lateral event continuity than Figure 8e,g. The PSNR value of model R3 in Table 3 is 20.18 dB, which is also the highest among other models, yielding that it can recover more reflectivity events at accurate locations.

Figure 9 denotes the crossplots between the two traces (x= 2400 and 8000 m) from the true band-limited reflectivity of the Foothills example and the ones predicted using model R1 (diamond scattered points) and model R3 (round scattered points) separately. Both traces indicate that model R3, using multiple reflections, can predict a higher correlation with the true band-limited reflectivity values than model R1. Since the relation between the prediction and true label should be linear, the slope of model R3 results (round scattered points in Figure 9) is closer to one compared with model R1 (diamond scattered points).

### 3.2. Example 2: Overthrust

A 2D subset of a synthetic 3D Overthrust model is the second numerical example, representing a more complicated geological structure with thin layers. The example size is 818 × 1602 points, with 79 shots simulated at the near surface. For further neural network prediction purposes, we extract 768 × 1536 points from the original example. Figure 10 represents the result for this Overthrust example. The smooth reflectivity is shown in Figure 10a. We can observe that the RTMM image (Figure 10d) yields extended subsurface illumination compared with the RTM image (Figure 10c). Correspondingly, model R3 prediction using smooth input and multiple energy in Figure 10f still provides better augmented resolution and accuracy than model R1 and R4 results (Figure 10e,g). The PSNR value of model R3 results in Table 4 is 24.61 dB, whereas model R1 and R4 results are 24.09 dB and 20.61 dB, respectively.

For the windowed example (red box in Figure 10h), Figure 11f displays enhanced information for the reverse fault from model R3 prediction in comparison with models R1 and R4 (Figure 11e,g). For example, the top layer above the small Overthrust at around 2500 m depth can be seen clearly with fined resolution in Figure 11f, whereas Figure 11e,g gives blurred and smoothed predictions. Furthermore, model R3 can recover the lateral variations with significant amplitude for the thin-layer structures below the reverse fault. The PSNR of the model R3 result is the highest value, 20.11 dB, shown in Table 4.

### 3.3. Example 3: SEAM Phase 1

The two examples above show our proposed neural network R3 can handle complex subsurface structures with faults and thin layers. We use another example to show model R3’s generalization ability on a more complicated model with thinner layers, folds, and a salt body. Figure 12 shows the SEAM Phase 1 velocity model. Note that SEAM Phase 1 has not been used in training or validating neural network workflow. We extracted a part of the original model with 801 × 1301 points. There are 35 shots and 250 receivers separated by 240 and 40 m, with a 15 Hz dominant Ricker wavelet. We chose sparse source–receiver coordination, because insufficient obtained data are normal in a real case. We want to explore the power of multiple reflection energy and neural network applied in this project to see if the pretrained neural network can classify useful multiple reflections from noisy data. The total time recording length is 7.2 s, with a 0.8 ms sampling rate. Since we resized this model to 768 × 1024 points to be fed into our pretrained neural network, the maximum offset is changed to 8.192 km. A smoothed background velocity is input to the reverse-time migration to avoid accurate information leakage.

Similar to previous examples, reverse-time migration after using multiple reflections (Figure 12d) gives a more accurate top layer structure than without using multiples (Figure 12c). Combined with PSNR comparison (Table 5), the model R3 prediction shown in Figure 12f improved reflectivity resolution and precision. Its PSNR is 26.32, which is larger than the model R1 prediction of 24.58.

We list a windowed example in Figure 13 for a detailed comparison. Model R3 result (Figure 13f) can indicate clearer events compared with model R1 prediction (Figure 13e). For example, in Figure 13e, model R1 result has a more blurred top layer with artifacts near the top above 800 m depth, whereas the prediction generated by model R3 at that area is clear with enhanced quality. For deeper events, model R3 also provides high resolution for dipping reflectors below 1500 m. Accordingly, the PSNR of this windowed example obtained by model R3 is 22.88, which is higher than the model R1 result.

Furthermore, the crossplot for this example, shown in Figure 14, proves that model R3 prediction (round scatter points) gives a higher correlation with the true band-limited reflectivity compared with model R1 prediction (diamond scatter points).

### 3.4. Discussion

***Input channels selection:*** We relied mostly on intuition and experimentation for deciding the input channels. For example, we compared results with a background velocity instead of a smooth reflectivity for the second input channel, but the results deteriorated. Although both results were alike, we observed more artifacts when using the background velocity, in particular at the boundaries between windows. This observation suggests that the network learns to calculate the directional derivative from the velocity in the second case, which accentuates the footprint caused by window overlapping. This example shows that often we can understand what the network does simply by experimenting with different inputs.

***True labels/output selection:*** The currently proposed method uses a band-limited reflectivity as the true label for training, generated from the training velocity models by converting to time, convolving with a wavelet, and converting back to depth. This is one of many possible choices. For example, we also tried to use the full bandwidth reflectivity with the expectation of extending the frequency band from the inputs when making the inference. The average spectrum for the output was undoubtedly more expansive than a band-limited label, but the results show many artifacts. We can explain this result by the network learning to perform deconvolution, which is sensitive to noise levels larger than the signal as we move away from the dominant frequencies. Once again, we can understand the network by experimenting with different inputs and outputs and conclude that the same constraints as classical processing limit the network. It calculates the operators by extracting the information from the data instead of hard-coded rules.

***U-Net architecture selection:*** For the U-Net architecture, we tried a shallower network as well, with five blocks of convolutional layers on the encoder and decoder parts, instead of the six blocks shown in Figure 3. The results were blurry for thin layers and small structures and not as good as the U-Net results in this paper. This indicates that small details extracted by the sixth block, shown at the bottom in Figure 3, were important for training and prediction. For the input size, we chose to use 256 × 256 points with two channels. We also tried other sizes, larger and smaller and square and rectangular, but the chosen size seemed to be optimal for this problem. For the encoder part, the filter shape changes from 16 to 512, and the kernel size decreases from 11 to 1 as the filter shape increases. There is no stride included in the convolutional layers. The padding is set to “same”, preserving the input size for each layer. The activation functions for all the convolutional layers were set to Rectified Linear Unit (ReLU). To initialize the layer weights, we use the “He” initializer [51], which draws samples from a truncated normal distribution centered on zero. For the downsampling, we use maxpooling with a 2 × 2 size. A batch normalization layer was applied after each activation function. Additionally, we combined “drop out” with batch normalization in each block to reduce generalization errors.

***Mean squared error (MSE) and mean absolute error (MAE):*** Mean squared error (MSE) penalizes larger prediction errors compared with mean absolute error (MAE), because significant errors are emphasized and have a relatively greater effect on the value of the performance metric. After testing with MSE and MAE, respectively, we found MAE results can be lower than MSE in workflows 2 and 4, whereas they perform worse when applied in workflows 1 and 3. The result indicates the MAE can handle well with accurate input, but it has limits on smooth input.

### 3.5. Model Performance and Computation Time

Figure 14 illustrates the model loss comparison between RTM-CNN and RTMM-CNN with fifty iterations when the starting learning rate is 0.001. Due to the true band-limited reflectivity input, model R4 (point dashed line) provides the lowest loss value. The model R3 (solid line) can converge to a smaller loss value, e.g., 0.0005, than the baseline model R1 (dashed line) for a smoothed input. For the validation loss shown in Figure 15, model R4 (point dashed line) gives the lowest loss value after 50 iterations, as it is used as a regularizer. Model R3 (solid line) can still converge to a smaller value than model R1 (dashed line), meaning model R3 works better in the validation set.

When training the network, each iteration takes around 53 s. We set the number of iterations to a large number (200), but training usually stops at around 50 iterations, which stops converging. Each neural network will take approximately 2650 s of runtime with an NVIDIA K80/T4 16GB GPU and 25.46GB RAM. The migration process is performed on an NVIDIA GeForce RTX 2080 Ti with 64 GB RAM. For the data preparation, each shot takes around one second for forward modeling and three seconds for imaging. Compared with the proposed method in this paper, LSRTM requires one forward modeling and one migration per iteration. The runtime for LSRTM can extend to several hours when the number of iterations is large. By contrast, the proposed method works directly with a single RTM calculation. The computational cost is reduced, because the cost of inference in neural networks is very small. Although the training can be computationally expensive, more than a regular LSRTM, this is performed only once. These were all simple 2D models, so an extension of this work to 3D would probably add one order of magnitude to this time.

## 4. Conclusions

The proposed RTMM-CNN method, which incorporates multiple energy in illumination, is capable of improving the quality of reflectivity obtained from migration, particularly when applied to a smooth initial model. The trained neural network takes advantage of multiple reflections and a reflectivity input from the background velocity model. The former enhances subsurface structure illumination, while the latter allows the neural network to accommodate for velocity errors. The network, trained with multiple reflections and a true velocity model, serves as a preconditioner that restricts the range of potential parameters due to the supplementary information it contains. Once a smoothed reflectivity is fed into the pretrained model, a new fine-tuned model can be obtained by further training to tolerate additional biases caused by preconditioning. The U-Net operator functions as an approximation of the inverse of the Hessian, suppressing image artifacts and enhancing the resolution of reflectors. This paper represents an initial step towards using multiple reflections for subsurface imaging with U-Net in realistic scenarios with limited velocity information. The neural network model exhibits robust generalization capabilities across diverse geology models. It is expected that the effectiveness of the neural network will be further improved with the emergence of even smoother migration velocity models.

## Figures and Tables

**Figure 1 sensors-23-04012-f001:**
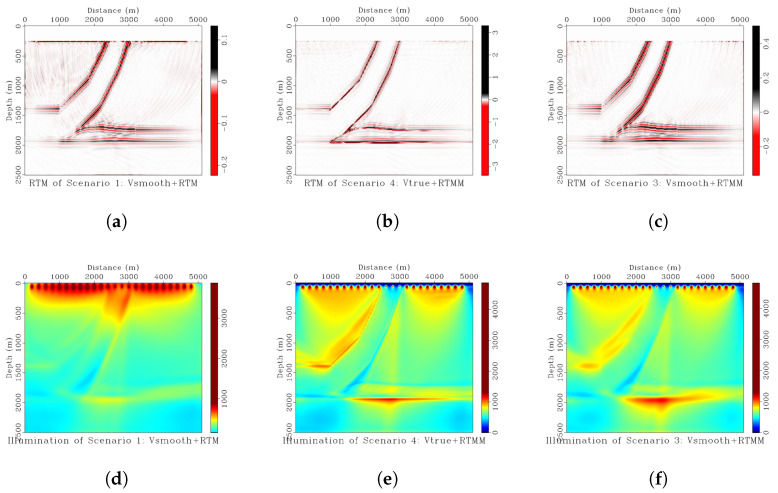
(**a**) RTM of scenario 1 using smoothed background velocity. (**b**) RTM of scenario 4 using true velocity as the input with multiple reflections. (**c**) RTM of scenario 3 using smoothed velocity with multiple energy. (**d**) Shot illumination of scenario 1. (**e**) Shot illumination of scenario 4. (**f**) Shot illumination of scenario 3.

**Figure 2 sensors-23-04012-f002:**
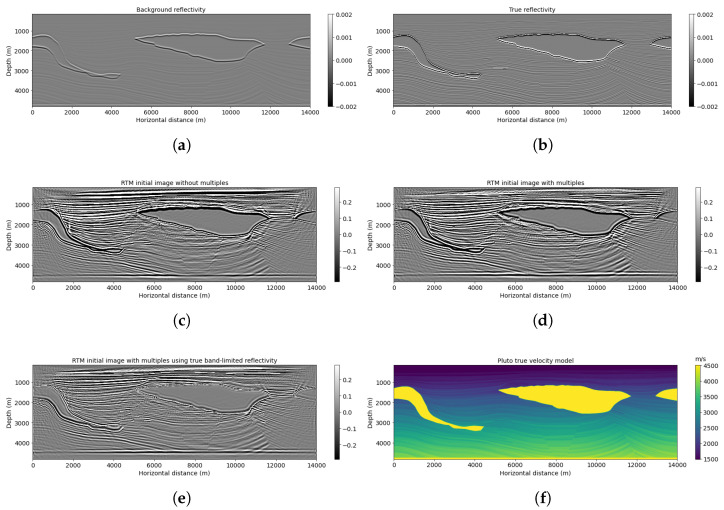
Pluto example: (**a**) background reflectivity, (**b**) true reflectivity, (**c**) RTM image without multiple energy, (**d**) RTM image with multiple energy, (**e**) RTMM with true band-limited reflectivity, and (**f**) true velocity model.

**Figure 3 sensors-23-04012-f003:**
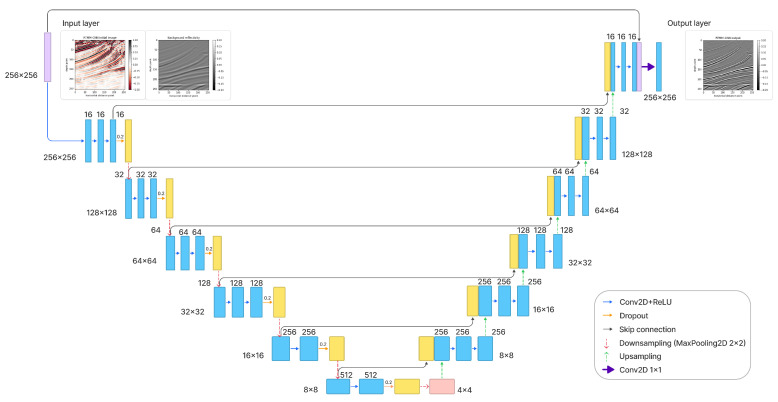
Detailed workflow of the U-net architecture. Each blue box represents a multichannel feature. The yellow boxes stand for the concatenated copied features from the encoder part. The arrows between boxes correspond to the different operations, as shown in the right legend. The number of channels is located on top of the box and the image dimensionality is denoted on the left or right edge.

**Figure 4 sensors-23-04012-f004:**
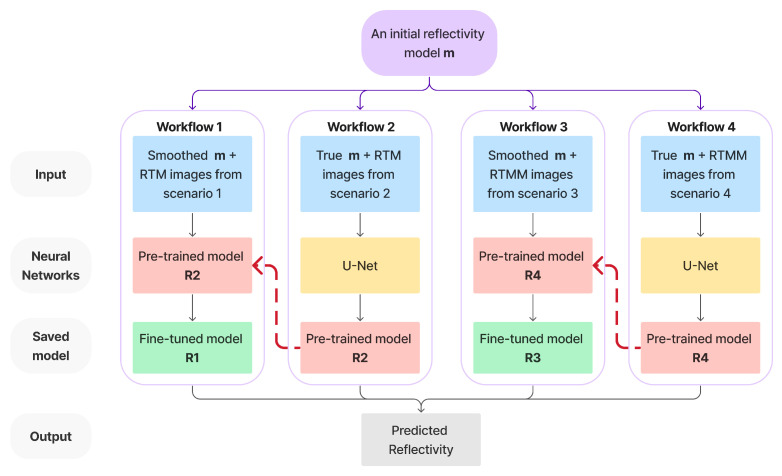
Neural network model plan for four scenarios.

**Figure 5 sensors-23-04012-f005:**
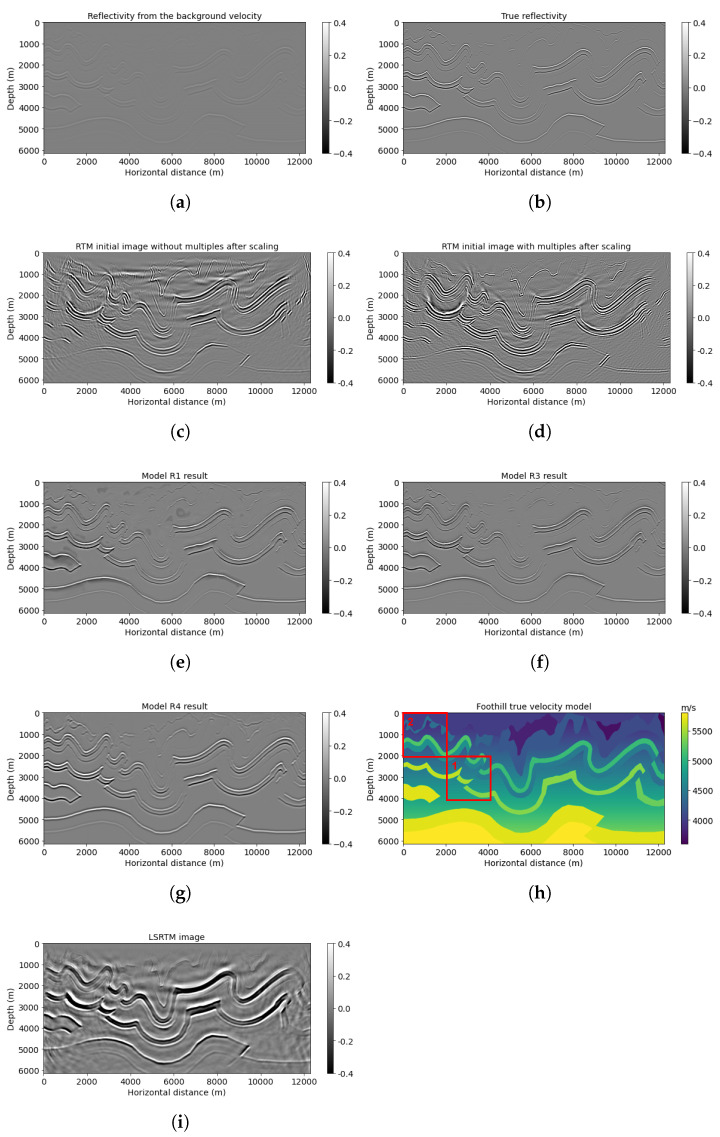
Canadian Foothills model results: (**a**) reflectivity from the background velocity, (**b**) true band-limited reflectivity, (**c**) RTM image without multiple reflections, (**d**) RTM image with multiple reflections, (**e**) model R1 result based on workflow 1, (**f**) model R3 result based on workflow 3, (**g**) model R4 result based on workflow 4, (**h**) true Foothills velocity, and (**i**) LSRTM result after 15 iterations. The boxes indicate areas shown in detail in the next figures.

**Figure 6 sensors-23-04012-f006:**
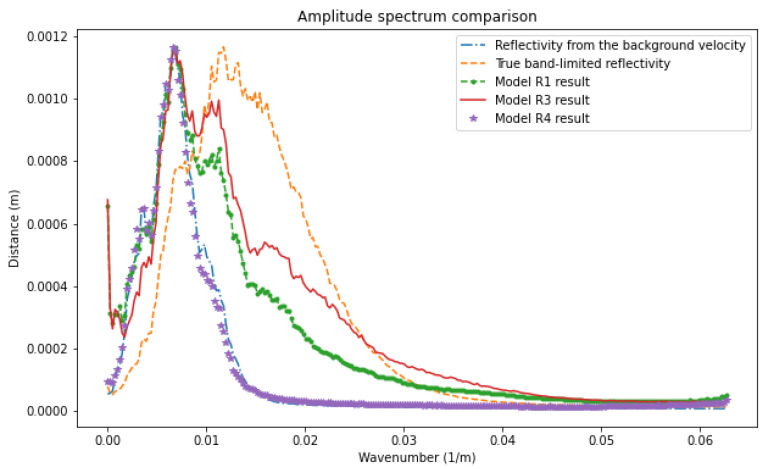
Amplitude spectrum comparison between models R1, R3, and R4 results for the Canadian Foothills example.

**Figure 7 sensors-23-04012-f007:**
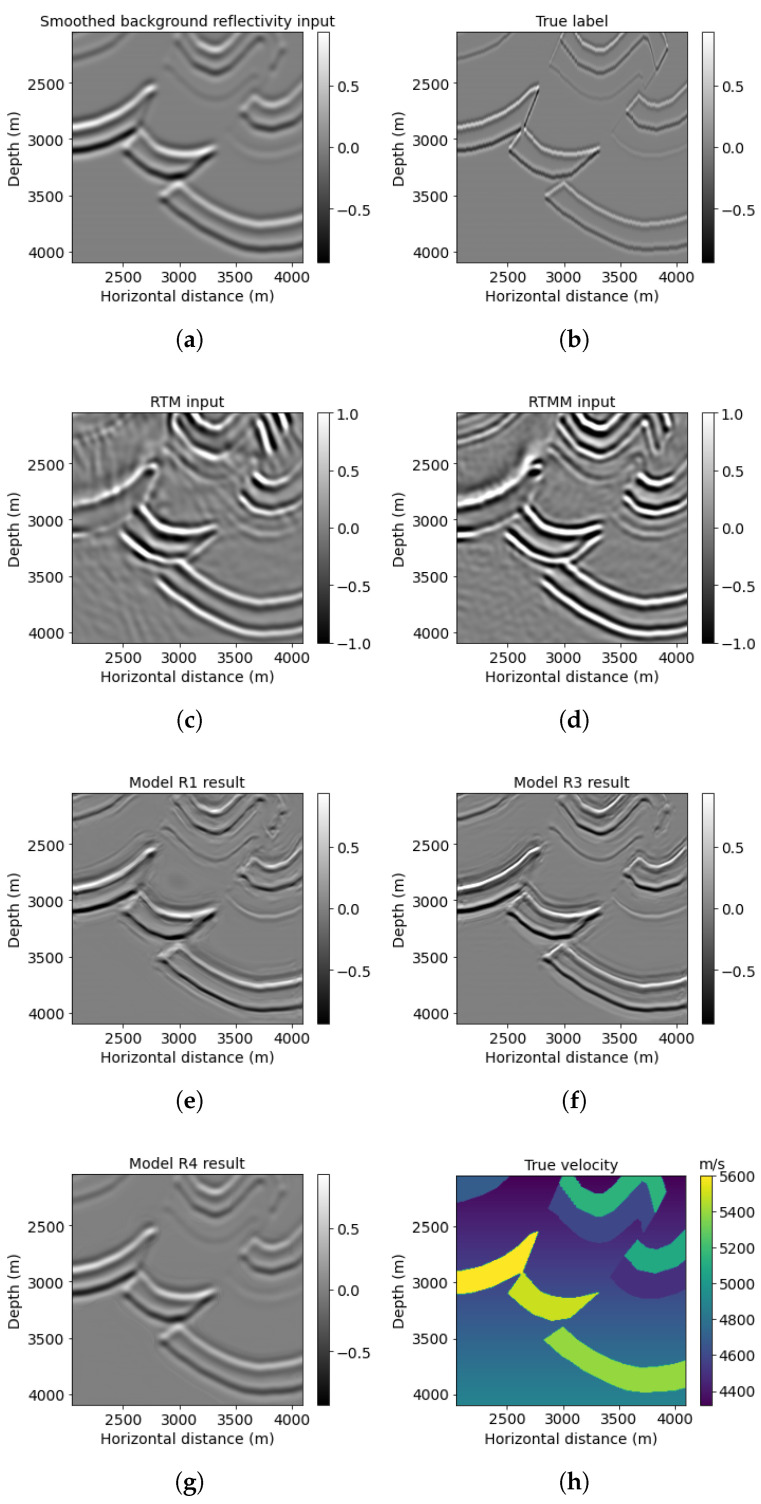
Foothills red box No. 1 results: (**a**) reflectivity from the background velocity, (**b**) true windowed band-limited reflectivity, (**c**) RTM image without multiple energy, (**d**) RTM image with multiple energy, (**e**) model R1 result based on workflow 1, (**f**) model R3 result based on workflow 3, (**g**) model R4 result based on workflow 4, and (**h**) true windowed velocity.

**Figure 8 sensors-23-04012-f008:**
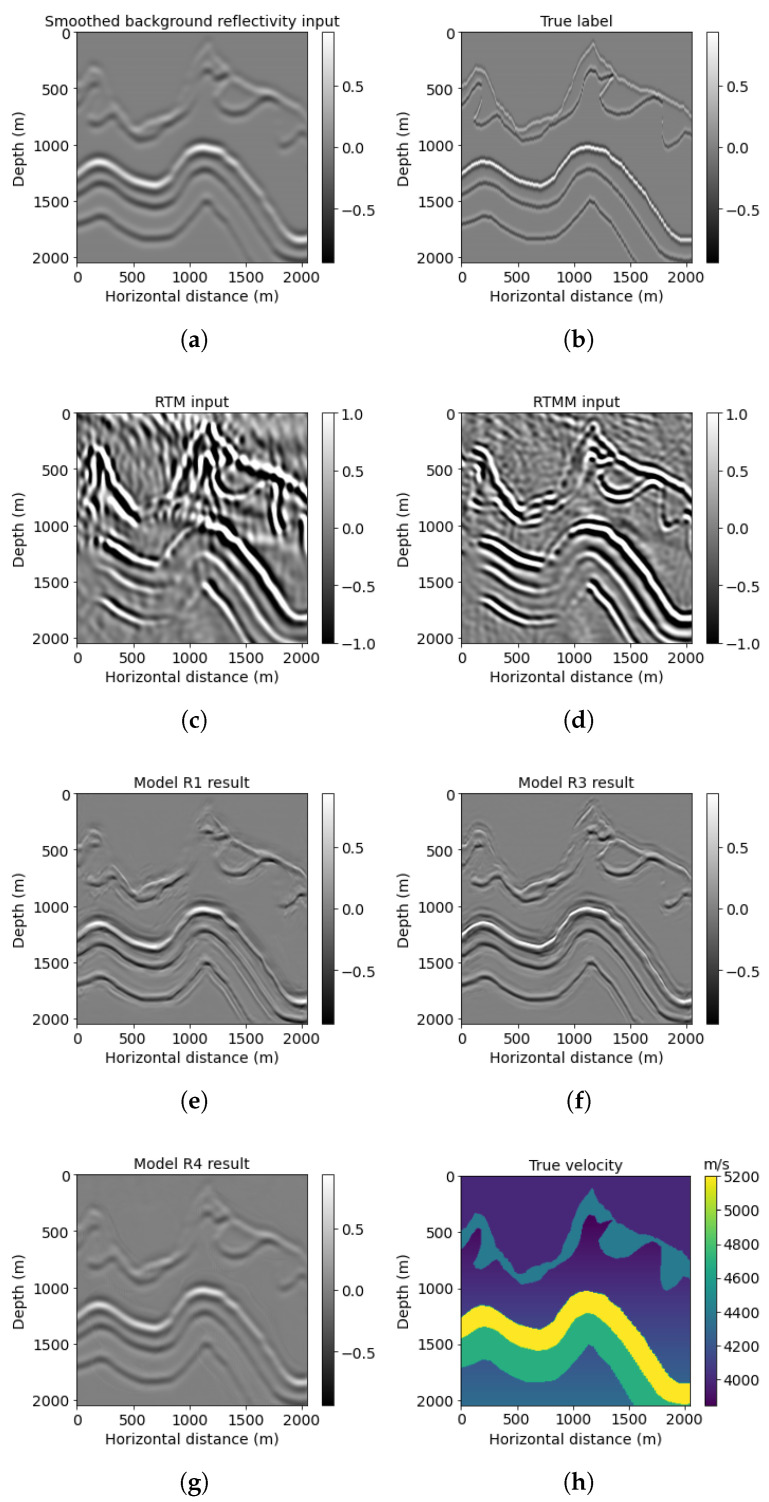
Foothills red box No. 2 results: (**a**) reflectivity from the background velocity, (**b**) true windowed band-limited reflectivity, (**c**) RTM image without multiple energy, (**d**) RTM image with multiple energy, (**e**) model R1 result based on workflow 1, (**f**) model R3 result based on workflow 3, (**g**) model R4 result based on workflow 4, and (**h**) true windowed velocity.

**Figure 9 sensors-23-04012-f009:**
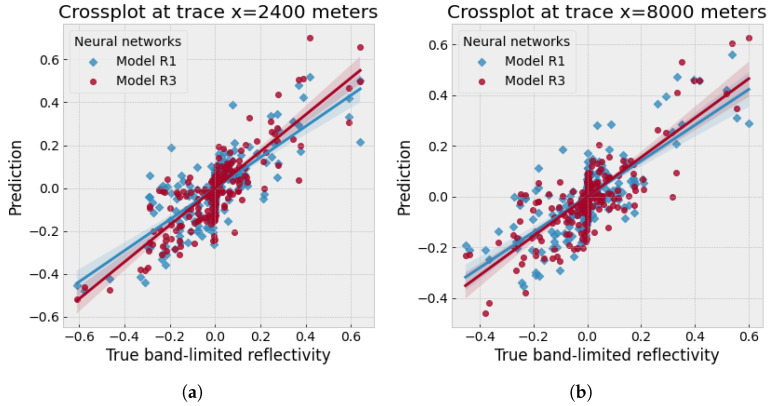
Crossplots for the Canadian Foothills example: the true band-limited reflectivity against the predicted reflectivity by using models R1 and R3, respectively.

**Figure 10 sensors-23-04012-f010:**
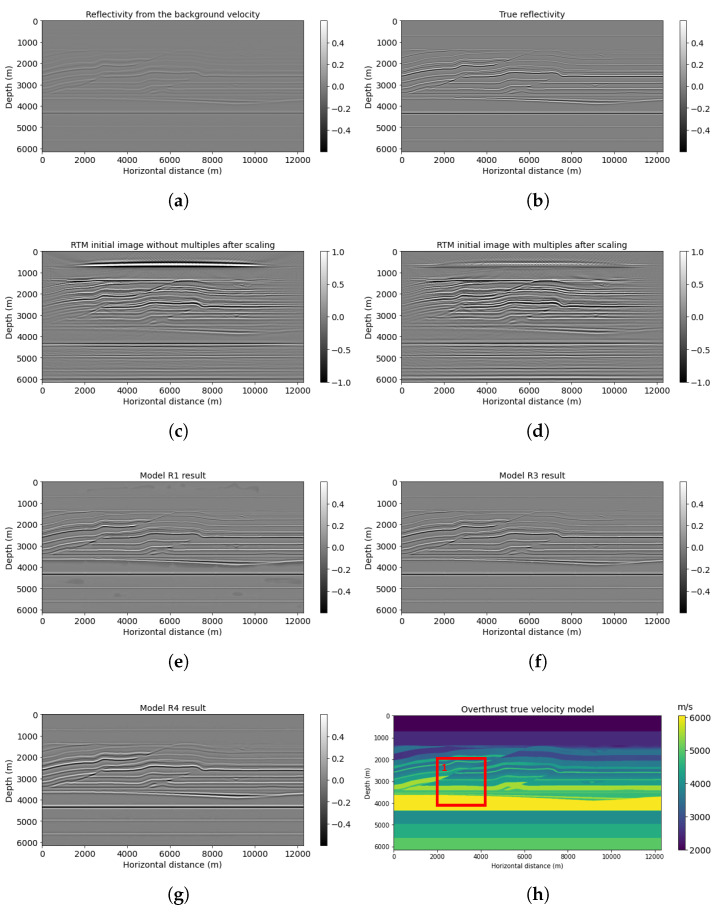
Overthrust model results: (**a**) reflectivity from the background velocity, (**b**) true band-limited reflectivity, (**c**) RTM image without multiple energy, (**d**) RTM image with multiple energy, (**e**) model R1 result based on workflow 1, (**f**) model R3 result based on workflow 3, (**g**) model R4 result based on workflow 4, and (**h**) true Overthrust velocity.

**Figure 11 sensors-23-04012-f011:**
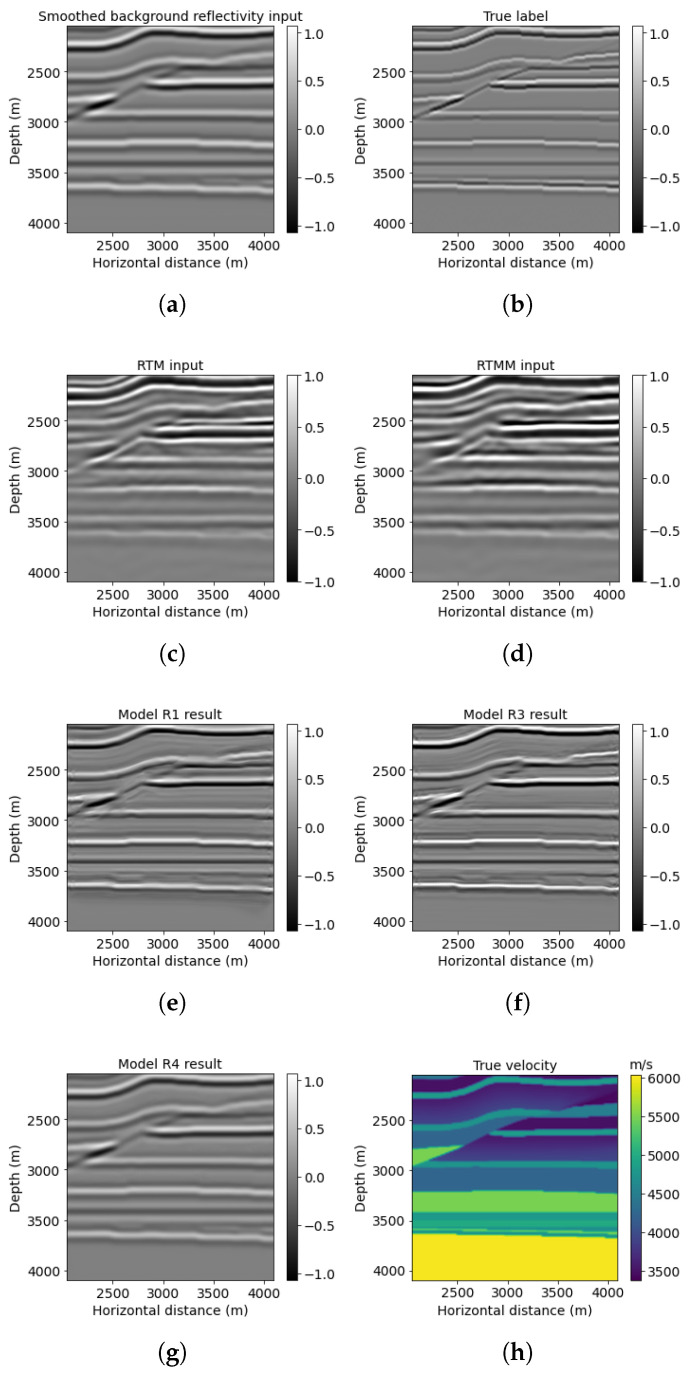
Overthrust red box results: (**a**) reflectivity from the background velocity, (**b**) true windowed band-limited reflectivity, (**c**) RTM image without multiple energy, (**d**) RTM image with multiple energy, (**e**) model R1 result based on workflow 1, (**f**) model R3 result based on workflow 3, (**g**) model R4 result based on workflow 4, and (**h**) true windowed velocity.

**Figure 12 sensors-23-04012-f012:**
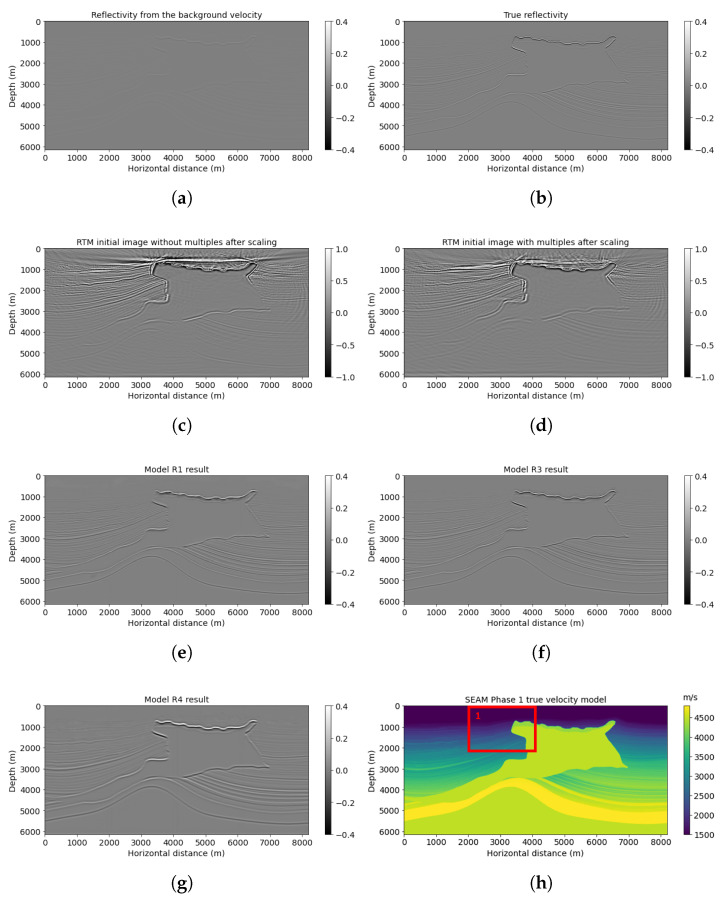
SEAM model results: (**a**) reflectivity from the background velocity, (**b**) true band-limited reflectivity, (**c**) RTM image without multiple energy, (**d**) RTM image with multiple energy, (**e**) model R1 result based on workflow 1, (**f**) model R3 result based on workflow 3, (**g**) model R4 result based on workflow 4, and (**h**) true SEAM velocity.

**Figure 13 sensors-23-04012-f013:**
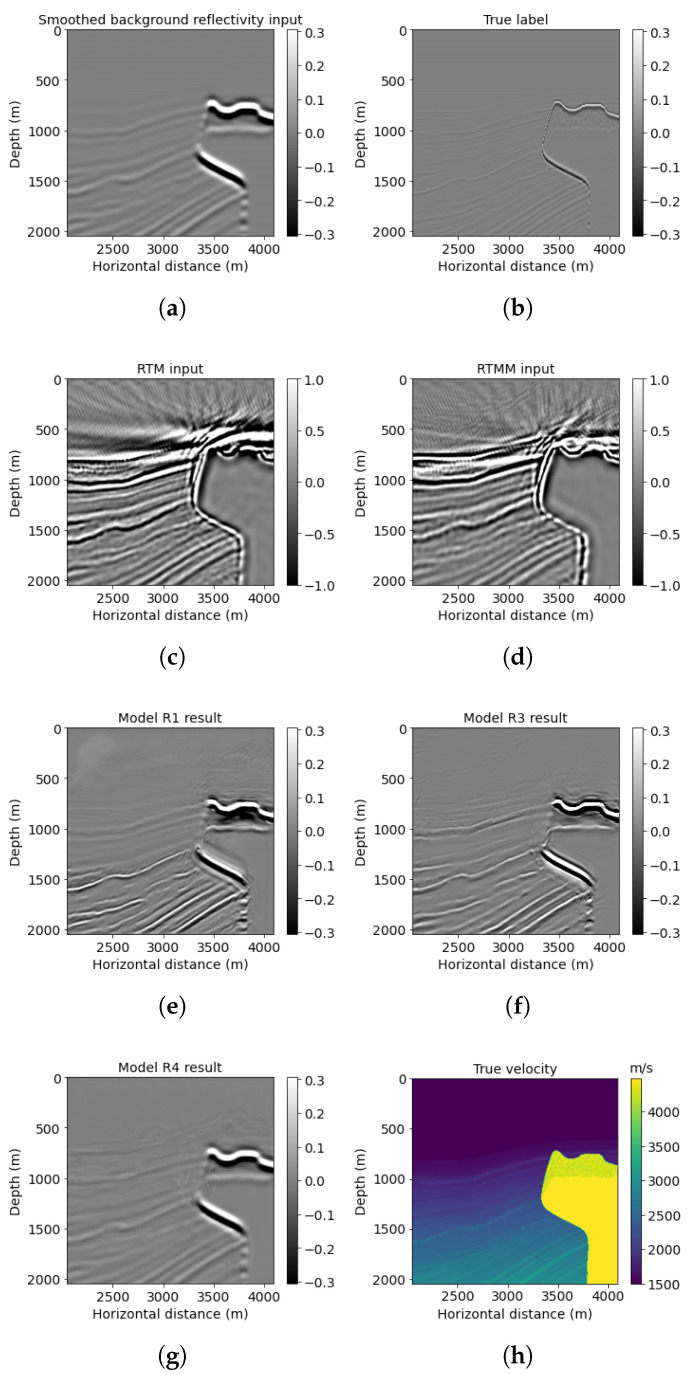
SEAM Phase 1 red box results: (**a**) reflectivity from the background velocity, (**b**) true windowed band-limited reflectivity, (**c**) RTM image without multiple energy, (**d**) RTM image with multiple energy, (**e**) model R1 result based on workflow 1, (**f**) model R3 result based on workflow 3, (**g**) model R4 result based on workflow 4, and (**h**) true windowed velocity.

**Figure 14 sensors-23-04012-f014:**
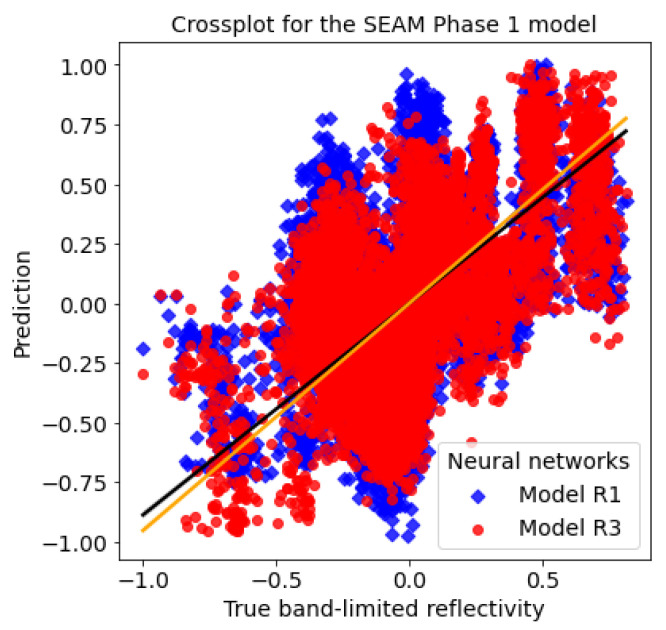
Crossplots for the SEAM Phase 1 example: the true band-limited reflectivity against the predicted reflectivity by using the model R1 and R3, respectively.

**Figure 15 sensors-23-04012-f015:**
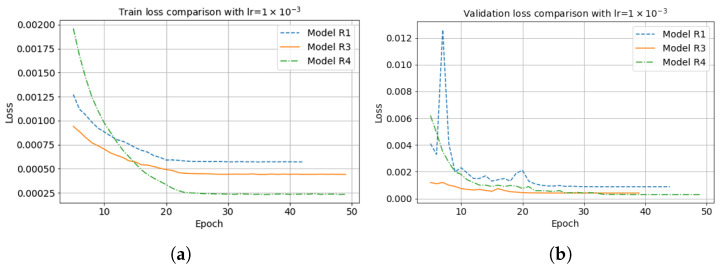
(**a**) Model train loss and (**b**) validation loss comparison between different neural network models with fifty iterations.

**Table 1 sensors-23-04012-t001:** U-net architecture encoding.

Layer Number	Type	Size	Output
1	Input		256 × 256 × 2
2	Conv2D	16 filters	256 × 256 × 16
3	Batch Normalization		256 × 256 × 16
4	Conv2D	16 filters	256 × 256 × 16
5	Batch Normalization		256 × 256 × 16
6	Conv2D	16 filters	256 × 256 × 16
7	Batch Normalization		256 × 256 × 16
8	Dropout	20%	256 × 256 × 16
9	MaxPooling2D	2 × 2	128 × 128 × 16
10	Conv2D	32 filters	128 × 128 × 32
11	Batch Normalization		128 × 128 × 32
12	Conv2D	32 filters	128 × 128 × 32
13	Batch Normalization		128 × 128 × 32
14	Conv2D	32 filters	128 × 128 × 32
15	Batch Normalization		128 × 128 × 32
16	Dropout	20%	128 × 128 × 32
17	MaxPooling2D	2 × 2	64 × 64 × 32
18	Conv2D	64 filters	64 × 64 × 64
19	Batch Normalization		64 × 64 × 64
20	Conv2D	64 filters	64 × 64 × 64
21	Batch Normalization		64 × 64 × 64
22	Conv2D	64 filters	64 × 64 × 64
23	Batch Normalization		64 × 64 × 64
24	Dropout	20%	64 × 64 × 64
25	MaxPooling2D	2 × 2	32 × 32 × 64
26	Conv2D	128 filters	32 × 32 × 128
27	Batch Normalization		32 × 32 × 128
28	Conv2D	128 filters	32 × 32 × 128
29	Batch Normalization		32 × 32 × 128
30	Conv2D	128 filters	32 × 32 × 128
31	Batch Normalization		32 × 32 × 128
32	Dropout	20%	32 × 32 × 128
33	MaxPooling2D	2 × 2	16 × 16 × 128
34	Conv2D	256 filters	16 × 16 × 256
35	Batch Normalization		16 × 16 × 256
36	Conv2D	256 filters	16 × 16 × 256
37	Batch Normalization		16 × 16 × 256
38	Dropout	20%	16 × 16 × 256
39	MaxPooling2D	2 × 2	8 × 8 × 256
40	Conv2D	512 filters	8 × 8 × 512
41	Batch Normalization		8 × 8 × 512
42	Conv2D	512 filters	8 × 8 × 512
43	Batch Normalization		8 × 8 × 512
44	Dropout	20%	8 × 8 × 512
45	MaxPooling2D	2 × 2	4 × 4 × 512

**Table 2 sensors-23-04012-t002:** U-net architecture decoding.

Layer Number	Type	Size	Output
1	Conv2D Transpose	256 filters	8 × 8 × 256
2	Batch Normalization		8 × 8 × 256
3	Concatenate		8 × 8 × 768
4	Conv2D Transpose	256 filters	8 × 8 × 256
5	Batch Normalization		8 × 8 × 256
6	Conv2D Transpose	256 filters	8 × 8 × 256
7	Batch Normalization		8 × 8 × 256
8	Conv2D Transpose	256 filters	16 × 16 × 256
9	Batch Normalization		16 × 16 × 256
10	Concatenate		16 × 16 × 512
11	Conv2D Transpose	256 filters	16 × 16 × 256
12	Batch Normalization		16 × 16 × 256
13	Conv2D Transpose	256 filters	16 × 16 × 256
14	Batch Normalization		16 × 16 × 256
15	Conv2D Transpose	128 filters	32 × 32 × 128
16	Batch Normalization		32 × 32 × 128
17	Concatenate		32 × 32 × 256
18	Conv2D Transpose	128 filters	32 × 32 × 128
19	Batch Normalization		32 × 32 × 128
20	Conv2D Transpose	128 filters	32 × 32 × 128
21	Batch Normalization		32 × 32 × 128
22	Conv2D Transpose	64 filters	64 × 64 × 64
23	Batch Normalization		64 × 64 × 64
24	Concatenate		64 × 64 × 128
25	Conv2D Transpose	64 filters	64 × 64 × 64
26	Batch Normalization		64 × 64 × 64
27	Conv2D Transpose	64 filters	64 × 64 × 64
28	Batch Normalization		64 × 64 × 64
29	Conv2D Transpose	32 filters	128 × 128 × 32
30	Batch Normalization		128 × 128 × 32
31	Concatenate		128 × 128 × 64
32	Conv2D Transpose	32 filters	128 × 128 × 32
33	Batch Normalization		128 × 128 × 32
34	Conv2D Transpose	32 filters	128 × 128 × 32
35	Batch Normalization		128 × 128 × 32
36	Conv2D Transpose	16 filters	256 × 256 × 16
37	Batch Normalization		256 × 256 × 16
38	Concatenate		256 × 256 × 32
39	Conv2D Transpose	16 filters	256 × 256 × 16
40	Batch Normalization		256 × 256 × 16
41	Conv2D Transpose	16 filters	256 × 256 × 16
42	Batch Normalization		256 × 256 × 16
43	Concatenate		256 × 256 × 16
44	Conv2D	1 filter	256 × 256 × 1

**Table 3 sensors-23-04012-t003:** PSNR (dB) comparison for Foothills example.

Prediction	Model R1	Model R3	Model R4
Total Foothills	24.84	**25.80**	20.59
Example 1	20.88	**22.58**	16.40
Example 2	19.16	**20.18**	17.03

**Table 4 sensors-23-04012-t004:** PSNR (dB) comparison for Overthrust example.

Prediction	Model R1	Model R3	Model R4
Total Overthrust	24.09	**24.61**	20.61
Example 1	19.06	**20.11**	15.99

**Table 5 sensors-23-04012-t005:** PSNR (dB) comparison for SEAM example.

Prediction	Model R1	Model R3	Model R4
Total SEAM	24.58	**26.32**	23.75
Example 1	21.52	**22.88**	20.14

## Data Availability

Available datasets can be found at https://drive.google.com/drive/folders/1ylL7vwwzGuApiHyLTrAFUepdqD3kxQOj?usp=sharing (accessed on 9 March 2023). Example data is licensed under the Creative Commons Attribution 4.0 International License.

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
