# Peer review of "Convolutional Neural-Network-Based Reverse-Time Migration with Multiple Reflections"

_sensors, 2023, doi:10.3390/s23084012_

Round 1

Reviewer 1 Report

Authors proposed a method based on a convolutional neural network (CNN) that behaves like a filter applying the inverse of the Hessian. This approach can learn patterns representing the relation between the reflectivity obtained through RTMM, and the true reflectivity obtained from velocity models through a residual U-Net with an identity mapping.

1. In the abstract section, I would suggest that the author should provide to the point and quantitative advantages of the proposed method.

2. The main contributions of this paper should be further summarized and clearly demonstrated.

3. The method/approach in the context of the proposed work should be written in detail.

4. In Line 236, "At the end of the encoding part, the image size is reduced from 256x256 to 4x4." How to reduce?

5. Some of Figures are not clear, please revise them.

6. The authors need to interpret the meanings of the variables.

7. One key background of this manuscript is the advanced techniques. Thus, the Introduction and/or related work section could be extended and incorporates additional discussions on the topics of advanced techniques, e.g., https://doi.org/10.3389/fendo.2022.1057089

https://doi.org/10.1088/1361-6501/ac9a61

https://doi.org/10.1016/j.asoc.2020.106724

Reviewer 2 Report

In this paper, the authors describe “Convolutional neural network-based reverse time migration with multiple energy”. It can become an interesting paper for Sensors after major revision. Followings are my comments.

(1)   Did the authors employ any data augmentation methods before training? If so, it should be mentioned.

(2)   What are the baseline models and benchmark results? For example, the authors compare the results with existing models by PSNR.

(3)   The Style of Eq. (10) is not in agreement with the journal style, and thus it requires a revision.

(4)   As the journal is printed in black and white, please make the different markers for the different results in Figs. 6 and 15.

Round 2

Reviewer 1 Report

This paper can be accepted now.

Reviewer 2 Report

Reviewer recommends to accept without comments.